# Evaluation of Fast Molecular Detection of Lymph Node Metastases in Prostate Cancer Patients Using One-Step Nucleic Acid Amplification (OSNA)

**DOI:** 10.3390/cancers13051117

**Published:** 2021-03-05

**Authors:** Svenja Engels, Lutz Brautmeier, Lena Reinhardt, Clara Wasylow, Friederike Hasselmann, Rolf P. Henke, Friedhelm Wawroschek, Alexander Winter

**Affiliations:** 1University Hospital for Urology, Klinikum Oldenburg, Department of Human Medicine, School of Medicine and Health Sciences, Carl von Ossietzky University Oldenburg, 26129 Oldenburg, Germany; engels.svenja@klinikum-oldenburg.de (S.E.); lutz.brautmeier@gmx.de (L.B.); lena@buecken-reinhardt.de (L.R.); cwasylow@gmx.net (C.W.); friederike.hasselmann@web.de (F.H.); wawroschek.friedhelm@klinikum-oldenburg.de (F.W.); 2Oldenburg Institute of Pathology, 26122 Oldenburg, Germany; r.p.henke@pathologie-oldenburg.de

**Keywords:** cytokeratin 19, metastases, OSNA, prostate cancer, sentinel lymph node

## Abstract

**Simple Summary:**

Prostate cancer (PCa) is the second most common cancer and one of the highestcauses of cancer deaths among men. The presence of lymph node metastases (LNM) is the strongest prognostic factor. Histological detection of LNM is the gold standard for LN staging. In clinical practice, only fractions of LNs are examined histopathologically, resulting in missed (micro-)metastases. Biomolecular techniques can enhance the identification of LNM but are not routinely used because of high cost. One-step nucleic acid amplification (OSNA) analyzes the entire LN by detecting cytokeratin 19 mRNA as a surrogate for LNM requiring less effort and allowing intraoperative application. To verify the reliability of OSNA for the first time in PCa, we examined LNs of patients undergoing prostatectomy and sentinel lymphadenectomy. Results were compared with histopathological and immunohistochemistry findings. OSNA identified LNM equivalent to, or in cases of micrometastases, even better than enhanced histopathology, and might be useful in intraoperative decision-making in personalized LN surgery.

**Abstract:**

*Background*: In clinical routine, only fractions of lymph nodes (LNs) are examined histopathologically, often resulting in missed (micro-)metastases and incorrect staging of prostate cancer (PCa). One-step nucleic acid amplification (OSNA) analyzes the entire LN by detecting cytokeratin 19 (CK19) mRNA as a surrogate for LN metastases requiring less effort than conventional biomolecular techniques. We aimed to evaluate performance of OSNA in detecting sentinel LN (SLN) metastases in PCa. *Methods*: SLNs (*n* = 534) of 64 intermediate- or high-risk PCa patients undergoing radical prostatectomy with extended and sentinel-guided lymphadenectomy were cut into slices and alternatingly assigned to OSNA and histopathology (hematoxylin-eosin staining, CK19, and CK AE1/AE3 immunohistochemistry). Sensitivity and specificity of OSNA and concordance and measure of agreement (Cohen’s kappa (κ)) between OSNA and histopathology were assessed. *Results:* Histopathology revealed metastases in 76 SLNs. Sensitivity and specificity of OSNA were 84.2% and 96.1%, respectively. Discordant results were recorded for 30 of 534 SLNs, revealing high concordance (94.4%). Twenty-four discordant cases were classified as micrometastases, indicating a possible allocation bias. In 18 cases, positive results were conferred only by OSNA resulting in seven LN-positive patients who were missed by histopathology. Overall, the level of agreement was high (κ = 0.78). *Conclusions:* OSNA provided a diagnosis that was as least as accurate as detailed histological examination and might improve LN staging in PCa.

## 1. Introduction

Prostate cancer (PCa) is the second most frequently diagnosed cancer and one of the most common causes of cancer deaths among men worldwide [1]. The presence of lymph node (LN) metastases (LNM) is the strongest prognostic factor for recurrence and cancer-specific mortality in PCa and occurs in 3–80% PCa patients undergoing radical prostatectomy [2]. Multimodal adjuvant treatment with either androgen-deprivation therapy alone or combined with pelvic radiation therapy has shown survival benefits in LN-positive PCa patients [3]. Despite advances in imaging, histological detection of LNM or pelvic LN dissection (LND) is still the gold standard for LN staging in clinically localized PCa [4]. Prostate-specific membrane antigen (PSMA) positron emission tomography (PSMA-PET) is clearly superior to computed tomography (CT) or magnetic resonance imaging (MRI) in LNM detection but is limited by its spatial resolution (2–5 mm) [5]. Recently, on the basis of a prospective multicenter study, a sensitivity of 40% for PSMA-PET was reported whereby the sensitivity was significantly lower in small-volume metastases (LN size > 10 mm: 68%; LN size < 10 mm: 30%) [6].

Because of the high proportion of small LNM, the manner of histological LN evaluation has a considerable influence on the number of LN-positive PCa patients identified. Cancer recurrence occurs in approximately 20% of postoperative LN-negative patients. One reason for this might be small LNM remaining undetectable by standard histopathology. The detection rate of LNM correlates with the number of sections examined [7]. Immunohistochemistry (IHC) and molecular analysis (e.g., real time polymerase chain reaction (RT-PCR)) can enhance the identification of occult metastasis or micrometastasis [8]. In clinical routine practice, histopathological LN evaluation is usually not extensively performed. The preparation of at least one section (3 to 4 mm slices in cases of larger LNs) in the longitudinal or transversal plane and hematoxylin and eosin (H&E) staining is still the standard in many centers [9]. However, because of the associated high expenditure and the limitations of standardization and automation, molecular techniques for LN evaluation are not routinely used in PCa.

To avoid these principle limitations and to enhance sensitivity, one-step nucleic acid amplification (OSNA) has been indicated as a fast, molecular intraoperative approach for the detection of LNM in routine breast cancer diagnostics [10,11,12,13]. Moreover, OSNA has already shown promising results in other tumor types (e.g., colorectal, gastric, endometrial, and lung cancer) [14,15,16,17]. This new biomolecular method exploits the fact that cytokeratin 19 (CK19) is expressed in the tissue of many different organs (e.g., prostate) but not normally in LNs, and is thus a surrogate for the presence of LNM [18]. OSNA quantifies the number of CK19 mRNA copies using reverse-transcription loop-mediated isothermal amplification (RT-LAMP) for gene amplification [19]. In conventional PCR, there is a risk of the amplification of nonspecific genomic deoxyribonucleic acid. The highly specific OSNA analysis eliminates this limitation using six primers and an isothermal reaction. Immunohistochemical studies revealed that CK19 expression frequently occurs in basal as well as in luminal cells of normal, dysplastic, and benign hyperplastic prostatic tissues and in PCa [20]. Using the OSNA assay, our group recently detected a high number of CK19 mRNA copies in all PCa specimens examined [21]. This suggests that OSNA may be suitable to identify LNM and to improve LN staging in PCa patients.

This prospective study aimed to analyze LN specimens of PCa patients with OSNA. To verify the reliability of this new molecular procedure, lymphatic tissue of intermediate- and high-risk PCa patients undergoing radical prostatectomy and sentinel pLND were examined using OSNA. Results were compared with conventional histopathological and IHC findings.

## 2. Materials and Methods

### 2.1. Patient Population

Sixty-five consecutive patients with intermediate- or high-risk PCa (PSA > 10 ng/mL and/or Gleason score ≥ 7) scheduled for radical retropubic prostatectomy with extended LN dissection (eLND) and magnetometer-guided sentinel LND (sLND) were recruited for this prospective study conducted between April 2017 and April 2018. Patients with prior hormone therapy or transurethral prostate surgery were excluded. The study was registered (DRKS00011401) and was ethically approved by the Medical Ethics Committee of the Carl von Ossietzky University Oldenburg (no. 2016-147). Informed and written consent was obtained from all patients.

### 2.2. Study Design

The study was designed according to the IDEAL-D recommendations as a two-step trial. In a first pilot trial, sentinel LNs (SLNs) from 17 patients were analyzed, and after initial evaluation of feasibility (IDEAL-D-Stage 2a), the cohort was enlarged in a second stage to a total of 65 patients (IDEAL-D-Stage 2b) [22].

Patients underwent eLND and magnetometer (SentiMag)-guided sLND after intraprostatic injection of superparamagnetic iron oxide nanoparticles (Sienna+; Endomagnetics Ltd., Cambridge, UK), followed by radical retropubic prostatectomy. Dissected SLNs were separately put on sterile compresses and immediately stored on ice. SLNs were measured, weighed, and intraoperatively cut into an even number of 2 mm-thick slices using new sterile disposable forceps and scalpels for each one to avoid RNA contamination from one SLN to another. Slices were alternatingly fixed by formalin for histopathology or frozen and stored at −80 °C for later OSNA analysis. Macroscopic suspicious LNs without magnetic activity were treated as SLNs to maximize the number of metastases analyzed by OSNA. The accuracy of OSNA and the measure of agreement (Cohen’s Kappa coefficient (κ)) were evaluated based on histological diagnosis.

### 2.3. OSNA

OSNA assay was performed using only designated instruments and reagent systems according to the manufacturer’s instructions (Sysmex Corporation, Kobe, Japan). Because of material boundaries, OSNA is limited in sample weight to a minimum of 50 mg and a maximum of 600 mg. LNs exceeding 600 mg were divided in smaller samples depending on LN size. To perform OSNA, samples were processed in several steps leading to the lysate as the final product. First, LN samples were homogenized in 4 mL LYNORHAG lysis buffer (Sysmex Europe GmbH, Norderstedt, Germany) for 60 s using LYNOPREP blades (Sysmex Europe GmbH) and a Polytron System PT1300D (Kinematica AG, Luzern, Switzerland) at 10,000 rpm. One milliliter of the resulting homogenate was centrifuged at 10,000× *g* for 1 min to remove cell debris. In the next step, 200 µL mid-section fluid (between the fat layer and the cell debris pellet) was pipetted twice into two small tubes, one for direct analysis and the other for storage at −80 °C for later use. The first lysate was further diluted 1:10 and 1:100 with LYNORHAG and then directly used for amplification without further RNA extraction or purification. The OSNA assay was implemented automatically using the Gene Amplification Detector RD-210 (Sysmex Europe GmbH), which has a capacity of 14 simultaneous analyses. Isothermal amplification reactions at 65 °C were performed with a ready-to-use reagent kit (LYNOAMP; Sysmex Europe GmbH) and the rise time (time needed for precipitation of magnesium pyrophosphate to reach a turbidity of 0.1 OD at 465 nm) and quantity of CK19 mRNA copies were determined using a daily implemented calibration curve. Using an established cutoff value for detecting LNM (>250 CK19 mRNA copies (c)/μL) [10] LN-samples were defined as “negative” (−, <250 c/μL) or “positive” (+) and categorized as micro- (250–4999 c/μL) or macrometastases (≥5000 c/μL). Figure 1a,c show typical amplification curves for macro- and micrometastasis, respectively.

### 2.4. Histopathology

The formalin-fixed LN-tissue was embedded in paraffin and cut into slices. After deparaffination and rehydration, slides were automatically stained with H&E. At least three 5 μm-thick sections from each of the original 2 mm-LN slices prepared during surgery were H&E stained and analyzed. One section of each 2 mm slice was examined with CK19-IHC staining. IHC was automatically performed with the Dako Envision Dual Link System-HRP (Dako Deutschland GmbH, Hamburg, Germany) by incubating peroxidase-blocked slides with CK19 antibody (A53-B/A2.26; medac GmbH, Wedel, Germany), labeling polymer, and 3,3′-Diaminobenzidine (DAB), and finally counterstaining the slides with H&E. Figure 1b,d show two examples of resulting CK19 IHC staining of a micro- and a macrometastasis, respectively. All sections were evaluated by one of three pathologists with high experience in urologic diseases. In cases of discordant results (OSNA positive/histologically negative), additional sections of the corresponding sample were stained with H&E and CK19 and checked with CK AE1/AE3.

For further investigation of cases with no or weak CK19 expression in LN metastases, CK19 IHC was also performed on tissue of primary tumors.

In order to confirm the accuracy of the OSNA results and the concordance of the OSNA and the histopathological method or to verify the reliability of the CK19 IHC in detection of metastasis, exemplary samples were additionally stained with CK AE1/AE3 IHC and an antibody against PSA.

### 2.5. RNA-Quality/Quantity

To assess the quantity and purity of the RNA, lysates of discordant samples underwent spectrophotometric analysis (Bio Photometer Plus 6132; Eppendorf, Hamburg, Germany) to obtain the total RNA concentration (ng/μL) and the 260:280 ratio, while automatic electrophoresis (Agilent 2100 Bioanalyzer System; Agilent Technologies, Santa Clara, CA, USA) was performed to obtain the total RNA concentration (ng/μL), the 28S/18S-ratio, and the RNA integrity number (RIN).

## 3. Results

In total, 1360 LNs were dissected from 65 patients. Nine LNs from one patient had to be excluded from the analysis because a prior hormone therapy became apparent at a later date. Of the 574 SLNs identified in the remaining 64 patients, 40 SLNs were excluded as they were too small to be cut and analyzed both by OSNA and histopathology. A summary of the patient and tumor characteristics is shown in Table 1.

Of the remaining 534 SLNs, 64 SLNs were determined positive both by OSNA and by histopathology as exemplarily shown in Figure 1 for one macro- and one micrometastasis, respectively. Twenty-four SLNs from 18 patients were determined positive by OSNA but not by histopathology. To review these discordant results, the pathologist examined additional sections of the respective samples including H&E-, CK19-, and for checking purposes in some cases AE1/AE3-staining and therefore could confirm the results of OSNA in six SLNs from five patients. Two of these patients had no other positive LNs, and thus they would have been staged as false-negatives in standard histopathology without the follow-up examination only performed under study conditions. So finally, 76 SLNs were determined to be metastasis-positive by conventional histopathology and/or IHC. The sensitivity of the OSNA assay was 84.2% and the specificity was 96.1%.

After follow-up examinations, there finally remained discordant results for 30 SLNs. For 18 SLNs from 13 patients a positive result was obtained solely by OSNA analysis but not by histological examination. Seven of these patients were staged as LN negatives by histopathology. In 12 SLNs from nine patients, histopathology identified a metastasis while OSNA gave a negative result. Four of these patients had no OSNA-positive LNs at all and therefore would have been missed using OSNA alone. Overall, the concordance was 94.4% and the level of agreement was high (κ = 0.78). Figure 2 shows an overview of the concordance between OSNA and histopathology considering the copies of CK19 mRNA detected by OSNA. In 24 of the 30 discordant results, metastases were classified as micrometastases. Excluding these cases assuming a high risk of tissue allocation bias, further analysis revealed 91.4% sensitivity, 100.0% specificity, 98.8% concordance, and κ = 0.95. In five of the six OSNA-negative macrometastases, IHC showed no or only occasional CK19-positive cells. A detailed view of the discordant cases is shown in Table 2 and Table 3.

Primary tumor tissue from one patient with two OSNA- and CK19 IHC-negative macrometastases showed also no expression in CK19 IHC. In primaries from patients presenting LNM with no or weak CK19 expression, CK19 IHC showed a heterogeneous pattern with partly also weak expression.

The additional staining performed with CK AE1/AE3 IHC and an antibody against PSA could confirm the results of CK19 IHC and thus the concordance of OSNA analysis and histopathological examination.

A quality check of the lysates showed high or sufficient (>100 ng/µL) RNA quality in 22 out of 25 samples examined. Two samples showed extremely low RNA quantity (>10 ng/µL), which might have affected the feasibility of OSNA.

## 4. Discussion

In the present study, the OSNA assay was used for the first time to detect CK19 expression in LN specimens of PCa patients. OSNA was demonstrated to have at least the same accuracy as intensive conventional histopathology in the detection of LNM. Application of OSNA may improve LN staging in PCa and reduce the workload of pathologists because most processes are automatic and can be performed by a skilled technician, and produce quantifiable results. Thus, there may be no difference in diagnostic accuracy for the diagnosis of LN metastasis between institutions.

In PCa, an accurate assessment of nodal status is crucial to calculate the risk of progression and to assign node-positive patients to appropriate adjuvant treatment with the opportunity of prolonging survival [23]. Whether a metastatic focus can be identified by conventional histopathological LN examination with several H&E-stained slides depends on the size of the metastatic lesion, its location within the LN, and the number of pathology slides. In PCa, conventional histopathology is currently considered as the gold standard for assessing LNs. However, consensus guidelines regarding the optimal procedure are lacking, despite the considerable consequences [24]. Commonly, only a few H&E-stained slides are examined for each node, and thus the LN tissue is only partially and in fact insufficiently examined. Metastatic lesions may be missed, resulting in an imprecise LN status. Although a more detailed histological examination of LNs, an increase in the number of histological slides, and performing IHC could provide more accurate LN staging, this would also enormously increase the pathological workload and costs, and would be impossible to undertake for all patients in routine clinical practice [7,25].

To overcome these issues, molecular methods such as RT-PCR that allow the examination of the entire LN have been developed to enhance sensitivity, especially for detecting small-volume tumor deposits or occult LN metastasis in PCa [8,26]. Several studies showed that recurrence can occur in patients in which the immunohistopathological LN examination was negative [27]. However, these molecular approaches are not yet applied to routine clinical practice because of their complexity and time-consuming nature. The OSNA assay using automated gene amplification permits a standardized and rapid procedure (approximately 30 min) that is suitable for intraoperative use at a reasonable cost. OSNA can detect metastatic foci regardless of their size or location by evaluating the whole node. In gastric cancer, OSNA identified metastatic lesions as small as 0.35 mm [28].

In view of the data presented in this study, the OSNA assay appears to be a promising tool for intraoperative LNM detection in PCa patients. This is supported by the recommendations for the clinical use of OSNA for intraoperative SLN diagnostics in breast cancer in several European countries and data available on other cancer types [14,16,17,29,30,31]. The accuracy of OSNA in PCa is similar to that of breast cancer, where a pooled analysis revealed a sensitivity of 89.3%, specificity of 94.8%, and a concordance rate of 93.8% [32].

OSNA analysis could provide the surgeon with the ability to choose intraoperatively whether to perform an additional eLND in PCa patients with positive SLNs. A systematic review showed a high diagnostic accuracy for sLND in detecting positive LNs in PCa [33]. In a consensus panel meeting, agreement was obtained on the statement that sLND should be combined with an eLND, at least in intermediate- and high-risk PCa patients, to resect additional LNM in non-SLNs [34]. However, the rate of complications rises along with the number of LNs removed. After OSNA analysis of the SLNs resected, in cases of negative OSNA results, LND could be personalized, sparing patients from unnecessary morbidity caused by eLND. This would imply that in such patients removal of additional LNs and conventional histopathological examination could be omitted. If one wants to proceed in this way, the results of our examinations on tissue from primary tumors from patients with low or weak CK19 expression in LN metastases would suggest that the CK19 expression should be checked first in the primaries. A similar approach has already been reported for colon and breast cancer [35,36].

Similarly to breast cancer, the total tumor load (TTL), defined as the amount of CK19 mRNA copy numbers in all positive SLNs (copies/µL), could also be a relevant prognostic factor in PCa. In breast cancer, a relationship between TTL and the presence of additional non-SLN metastases and correlation with a higher risk of recurrence could be shown [37,38,39]. In the future, the TTL could also be used in PCa and, for example, improve the management of adjuvant therapy. However, to clarify this perspective, further studies are needed in which the tissue of SLNs is fully examined by OSNA in analogy to breast cancer.

Consideration of TLT or pooling of nodes for OSNA analysis may also enable to reduce the costs associated with the use of this molecular approach in PCa. Taking into account multiple SLNs (median 8 SLNs in the present study) and costs for OSNA of about 80–90 Euro per node or sample, pooling of several SLNs into one microcentrifuge tube, as already reported for colon cancer, might reduce the number of molecular tests needed per patient and could help to integrate the OSNA procedure in daily practice [35,40]. However, it should be noted that SLNs in PCa can be relatively large and a sample for OSNA analysis must not exceed a weight of 600 mg.

Our detailed analysis of discordant cases indicates that the main factor influencing our results is the possible tissue allocation bias. Considering that one half of the LN is subjected to OSNA analysis and the other to histopathology/IHC, discordant results cannot be avoided because metastasis may be located in only one part of the sample. The uneven localization of metastatic foci inevitably caused discrepant results between these two modalities. For micrometastases accounting for 80% of the discordant cases, this risk is especially high. In fact, the use of alternate sections generally seems to be the main reason for discrepant results between molecular and histological analyses of SLNs [41].

Five of the six macrometastases not detected by OSNA showed no or only isolated/weak CK19 expression by IHC. In total, no/low CD19 expression was detected in six metastases originating from four patients. All of them met at least three of the following four prognostically poor criteria: pT3, high tumor volume, ISUP grade ≥3, poor Gleason grading including high grade 4 percentage, or tertiary grade 5 pattern. Therefore, it seems that in some cases, advanced and poorly differentiated tumors might show a lack of CK19 expression, so that metastases are not detectable by OSNA. However, in our pilot study analyzing primary PCa specimens, CK19 mRNA could be detected by OSNA in all cases, regardless of the degree of tissue differentiation [21]. Further investigations are needed to clarify this discrepancy.

A common limitation of molecular LN staging is that morphological information (e.g., metastasis size and extracapsular extension) cannot be obtained. However, CK19 mRNA transcript quantification by OSNA can differentiate between micro- and macrometastases, shows increased sensitivity in comparison with standard histopathology, and overcomes the issue of histopathology-associated errors on LN slicing or failure to detect small foci on microscopic examination. Furthermore, the quantification of CK19 mRNA reveals new perspectives. In breast cancer, the total tumor burden of the resected SLNs/total number of CK19 mRNA copies measured by OSNA (TTL) could be established as an independent predictor and integrated into nomograms predicting metastatic invasion of non-SLNs [42]. A correlation between the total number of CK19 mRNA copies in SLNs and distant disease-free and cancer-specific survival could be demonstrated, and risk groups defined [43].

Despite its strengths, our study has also some limitations beyond the principle issues of the molecular LN analysis mentioned above. Notably, a direct comparison with the results of other molecular techniques remains to be completed. However, we have recently initiated a study investigating LN specimens of PCa patients that includes detailed conventional histopathological examination as well as additional molecular analyses as a reference.

## 5. Conclusions

We demonstrate for the first time the ability of the molecular OSNA approach, which analyze the entire LN to detect LNM in PCa. OSNA identified metastases at an equivalent or, in cases of micrometastases, better rate than enhanced histological examination. Enabling reliable intraoperative diagnostics or personalized LN surgery based on the metastatic status of SLN, OSNA might thus improve LN staging in PCa. However, further studies are needed to verify discordant results and to compare OSNA with other biomolecular methods.

## Figures and Tables

**Figure 1 cancers-13-01117-f001:**
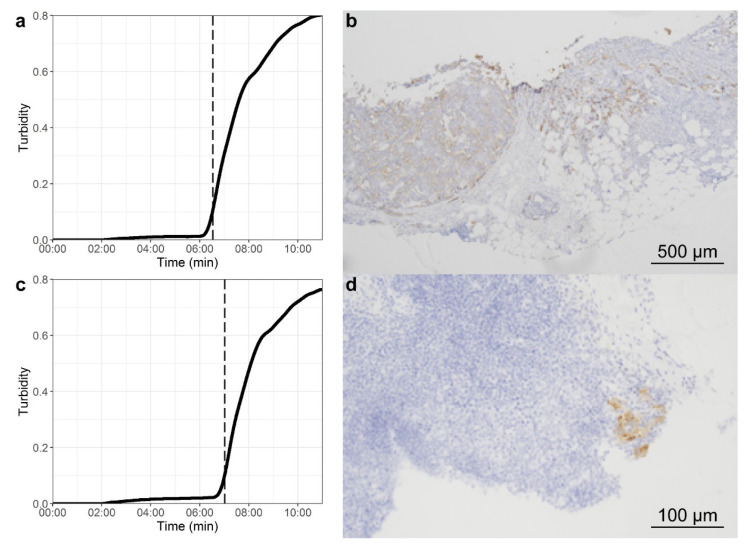
Detection of CK19 expression in one LN with macrometastasis (**a**,**b**) and in one LN with micrometastasis (**c**,**d**), each from the same PCa patient. (**a**,**c**) Amplification curves from OSNA analysis, which determines CK19 mRNA copy numbers (micrometastasis: 250–4999 c/μL, macrometastasis: ≥5000 c/μL) as a surrogate for metastasis. Vertical dotted lines plotted in the amplification curves express the rise time (time needed for precipitation of magnesium pyrophosphate to reach a turbidity of 0.1 OD at 465 nm). Rise time is shorter in macro- (06:32 min, (**a**)) than in micrometastasis (07:01 min, (**c**)). (**b**,**d**) Images show CK19 IHC stainings (total magnification for macrometastasis (**b**) is 400× and for micrometastasis (**d**) 2000×).

**Figure 2 cancers-13-01117-f002:**
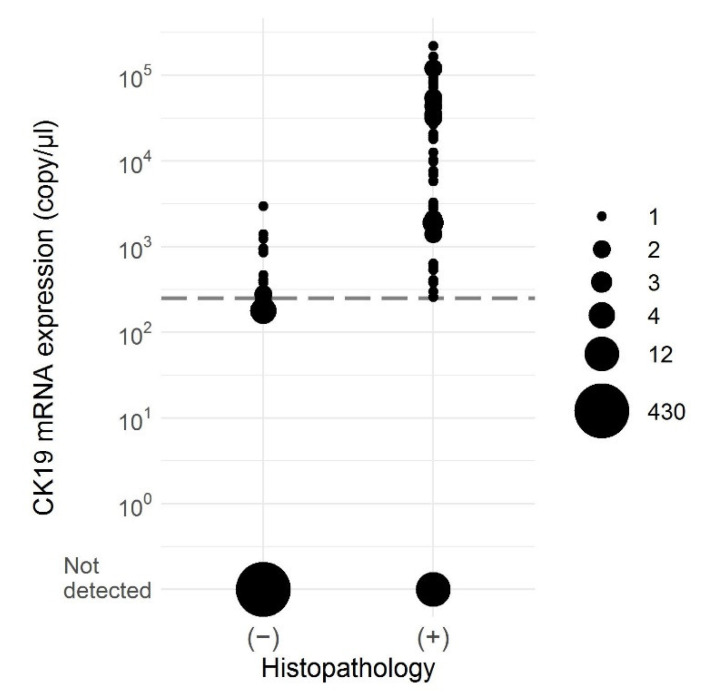
CK19 mRNA expression in SLNs (*n* = 534) revealed by OSNA analysis according to histopathological examination. The dotted line depicts the 250 copies/µL cut off value.

**Table 1 cancers-13-01117-t001:** Patient characteristics.

Characteristic	Overall *n* = 64	Patients with Pathological Negative LNs *n* = 41 (64%)	Patients with Pathological Positive LNs *n* = 23 (36%)
Age	69 (65.5–73)	69 (63.5–73.5)	69 (67–73)
Total PSA, ng/ mL	11.3 (9.3–17.8)	10.7 (9.0–14.2)	14.9 (9.9–22.3)
No. of LN removed	17 (13–22)	17 (13–22)	15 (13–20)
No. of SLN removed	8 (5–12)	9 (5–13)	8 (5–11)
No. of positive LN			3 (1–8)
Tumor stage (%)			
cT1c	22 (34.4)	21 (51.2)	1 (4.4)
cT2a	18 (28.1)	10 (24.4)	8 (34.8)
cT2b	3 (4.7)	2 (4.9)	1 (4.4)
cT2c	16 (25.0)	7 (17.1)	9 (39.1)
cT3	5 (7.8)	1 (2.4)	4 (17.4)
Biopsy Gleason score (%)			
6 (3 + 3)	1 (1.6)	1 (2.4)	0
7 (3 + 4)	30 (46.9)	25 (61.0)	5 (21.7)
7 (4 + 3)	10 (15.6)	8 (19.5)	2 (8.7)
≥8	23 (35.9)	7 (17.1)	16 (69.6)
Postoperative Gleason score (%)			
6 (3 + 3)	0	0	0
7 (3 + 4)	25 (39.1)	24 (58.5)	1 (4.4)
7 (4 + 3)	19 (29.7)	11 (26.8)	8 (34.8)
≥8	20 (31.3)	6 (14.6)	14 (60.9)
Pathologic stage (%)			
pT2	23 (35.9)	23 (56.1)	0
pT3a	12 (18.8)	11 (26.8)	1 (4.4)
pT3b	29 (45.3)	7 (17.1)	22 (95.7)
pT4	0	0	0

Data are given as median (interquartile range) or frequency (percentage). LN = lymph node, PSA = prostate specific antigen.

**Table 2 cancers-13-01117-t002:** Detailed view of OSNA positive/histopathological negative lymph nodes.

LN No.	OSNA *	Micrometastasis **	Histopathology	CK19 IHC
3-1	+	+	−	−
3-2	+	+	−	−
3-4	+	+	−	−
11-7	+	+	−	−
14-1	+	+	−	−
14-3	+	+	−	−
15-5	+	+	−	−
15-21	+	+	−	−
17-10	+	+	−	−
28-8	+	+	−	−
37-2	+	+	−	−
42-4	+	+	−	−
45-11	+	+	−	−
45-19	+	+	−	−
52-17	+	+	−	−
54-4	+	+	−	−
62-3	+	+	−	−
63-2	+	+	−	−

LN = lymph node; IHC = immunhistochemistry; *+: >250 CK19 mRNA copies/µL; **+: 250–4999 CK19 mRNA copies/µL; −: negative result.

**Table 3 cancers-13-01117-t003:** Detailed view of OSNA negative/ histopathological positive lymph nodes.

LN No.	OSNA *	Histopathology	CK19 IHC	Comments
1-1	−	macrometastasis	−	
1-3	−	macrometastasis	−	
15-15	−	−	+	macrometastasis only found in LN slices for IHC
44-1	−	macrometastasis	+	only single CK19 positive cells
64-6	−	macrometastasis	+	only single CK19 positive cells
64-11	−	macrometastasis	+	in parts faint CK19 positive cells
19-22	−	micrometastasis	+	
28-2	−	micrometastasis	+	only single CK19 positive cells
31-13	−	micrometastasis	+	
43-1	−	micrometastasis	+	
62-9	−	−	+	micrometastasis only found in LN slices for IHC
64-1	−	micrometastasis	−	

LN = lymph node; IHC = immunhistochemistry; *−: <250 CK19 mRNA copies/µL; −: CK19 negative result; +: CK19 positive result

## Data Availability

The data presented in this study are available on request from the corresponding author.

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
