# Peer review of "Evaluation of Fast Molecular Detection of Lymph Node Metastases in Prostate Cancer Patients Using One-Step Nucleic Acid Amplification (OSNA)"

_cancers, 2021, doi:10.3390/cancers13051117_

Round 1

Reviewer 1 Report

This is an interesting article describing the accordance between pathological examination and OSNA in detecting lymph node metastases from prostate carcinoma.

Overall, the article is well written. I think the authors could improve the "Results" section by adding a figure, to better illustrate the accordance between the two examinations.

In Table 1, please clarify if the patients are classified in positive or negative according to pathological or OSNA examination.

Author Response

Dear Reviewer

Thank you very much for your comments and the time you afforded to read and review our manuscript. Enclosed you will find our answers to your valuable suggestions.

Sincerely

Alexander Winter

Reviewer 2 Report

Engels et al. performed a prospective study to evaluate one-step nucleic acid amplification (OSNA) in sentinel lymph nodes in 65 prostate cancer cases.

This is an interesting study and a well written manuscript. I have some suggestions that should be addressed by the authors.

Methods: The authors should describe the processing of the SLNs in the fresh state more in detail. E.g.

How did the authors avoid RNA contamination from one LN to another when more than one SLNs were identified?

Results:

An evaluation on a case basis would be interesting for the reader. How often was OSNA the only method to identify a positive LN?

Total tumor load is a marker that is prognostic in breast cancer. Does TTL in prostate cancer correlate with metastases size or other established histopathologic prognostic factors? Is there a correlation with the risk of metastases in non-SLN?

Table 1: Why do the authors report T- and pT-stages separately?

CK-19 immunohistochemistry of the primary in the cases where the lymph node metastases showed no or weak expression? In cases where the primary is negative, OSNA evaluation could be contraindicated.

Discussion: costs: the authors should discuss the issue of costs since SLN mapping in prostate cancer always reveals several SLNs, and OSNA evaluation is relatively expensive. Could pooling (REF: Rakislova et al. 2017)

The authors should offer a model for OSNA testing in PC in routine. Would they recommend combining with conventional histology/IHC or would the count only on OSNA:

Author Response

(The authors gave the same response as above.)

Reviewer 3 Report

In the current clinical routine, PCa metastases in LNs are only examined by histopathological methods that leaves possibilities of under-detection of the lymph nodes metastases (LNM) due to various reasons. In the current study, authors were trying to tackle this by introducing one-step nucleic acid amplification (OSNA) on CK19 mRNA expression as a new detection modality. The results showed the two detection methods are in high concordance, but, there were discordance due to inevitable factors, such as the number of biopsies and the location of the LNM.

The results are very encouraging that more modality can be introduced into the clinical practice to improve the LNM detection. But, there are some concerns need to be addressed.

  • Can authors show some OSNA CK19 mRNA raw data, such as, amplification curve or cycle number, and some staining images of CK19 to show the concordance and discordance of the detection modalities?

  • Staining of CK19 is not the best way of identifying LNM in patients’ biopsies. In order to confirm the accuracy of the OSNA results, and the concordance of the OSNA and the histopathological method, more IHC targets should be applied to confirm the LNM in the biopsies. Please show at least one of following targets, NKX3.1, PSA or PSMA by IHC.

  • As the authors described, that there were 6 LNM detected only by IHC and 2 of them would have been missed if detected only by OSNA. Please justify the proposal of using OSNA as a surrogate of the current gold standard histopathological method. 

Author Response

(The authors gave the same response as above.)

Reviewer 4 Report

This  work addresses a timely and relevant clinical issue. The current submission is a continuation of previously published work, where the CK19 expression in prostate cancer tissue was examined. Authors evaluated here the CK19 status by OSNA method  within the 534 SLN from 64 patients, which is a representative cohort. The OSNA experiments were properly performed  with the use of dedicated instrument and reagents and in accordance with manufacturer instructions. This study considered 250 copies of cytokeratin 19 by μl as a criterion for positivity, which is the gold standard when working with OSNA. Next, the routine histopathological evaluation was performed to assess the correlation between the two methods. The findings are of great importance, however, I have a few additional questions for the authors.

1.OSNA as rapid intraoperative detection of SLN micrometastases was originally used in breast cancer. Is OSNA used in routine breast cancer diagnostics? Please provide more recent references. Please consider to discuss shortly this issue and elaborate eventually with recent guidelines.

2.The optimal approach for SLN mapping appears to be pathologic ultrastaging, based on multiple sections stained with hematoxylin/eosin and immunohistochemistry. Was the H&E staining performed and evaluated in this study? Please elaborate.

3.Was the CK19 expression evaluated by the histopathologist? How many experts evaluated the samples? This information is missing in the manuscript.

  1. How many slices from one SLN were IHC stained?
  2. How is the accuracy (sensitivity and specificity )of SLN from PCa testing comparing to breast cancer? Please consider broaden Discussion about this issue.
  3. Introduction, lines: 75-77. Reference regarding the use of OSNA in gastric cancer is missing.

Author Response

(The authors gave the same response as above.)

Round 2

Reviewer 2 Report

The authors addressed all points of the reviewer appropriately. In my opinion, the manuscript is suitable for publication, now.